# Photoacoustic Flow Cytometry Using Functionalized Microspheres for Selective Detection of Bacteria

**DOI:** 10.3390/mi14030573

**Published:** 2023-02-28

**Authors:** Robert H. Edgar, Anie-Pier Samson, Tori Kocsis, John A. Viator

**Affiliations:** 1Department of Bioengineering, University of Pittsburgh, Pittsburgh, PA 15260, USA; 2Department of Engineering, Duquesne University, Pittsburgh, PA 15282, USA

**Keywords:** acoustic, biosensing, optoacoustics

## Abstract

Photoacoustic flow cytometry is a method to detect rare analytes in fluids. We developed photoacoustic flow cytometry to detect pathological cells in body fluids, such as circulating tumor cells or bacteria in blood. In order to induce specific optical absorption in bacteria, we use modified bacteriophage that precisely target bacterial species or subspecies for rapid identification. In order to reduce detection variability and to halt the lytic lifescycle that results in lysis of the bacteria, we attached dyed latex microspheres to the tail fibers of bacteriophage that retained the bacterial recognition binding sites. We tested these microsphere complexes using *Salmonella enterica* (*Salmonella*) and *Escherichia coli* (*E. coli*) bacteria and found robust and specific detection of targeted bacteria. In our work we used LT2, a strain of *Salmonella*, against K12, a strain of *E. coli*. Using Det7, a bacteriophage that binds to LT2 and not to K12, we detected an average of 109.3±9.0 of LT2 versus 2.0±1.7 of K12 using red microspheres and 86.7±13.2 of LT2 versus 0.3±0.6 of K12 using blue microspheres. These results confirmed our ability to selectively detect bacterial species using photoacoustic flow cytometry.

## 1. Introduction

Bacterial contamination is part of all medical and industrial services [1,2]. Detection of bacterial contamination early and accurately is imperative to reduce negative impacts on patients, production, and general well-being. The two main areas of bacterial diagnostics occur in the manufacturing and food industry and in the medical field [3,4]. In both sectors, rapid determination of bacterial type and antibiotic sensitivities are required.

Rapid identification of bacterial infection is a pressing need in clinical care [5]. It is only after identification of the pathological agent that virulence, antibiotic resistance, and other relevant factors can be considered when determining optimal therapy [6]. Misdiagnosis can result in delayed therapy that can cause sepsis, multiple organ failure, and possibly death. Evaluating patients suspected of bacterial infection is a complex process with unique aspects of each case that may confound proper diagnosis. A system that can immediately identify the bacterial pathogen will result in better outcomes for millions of patients each year.

For identification of bacteria, culturing directly from a patient sample is the gold standard whereby all other detection methods are judged. Culturing of bacteria stems from at least 7000 B.C. with the production of cheese [7] and other fermented foods. As human beings developed various methods of food fermentation and preservation, so did their understanding of the effects of bacteria upon food production and preventing spoilage. Although people producing cheese and yogurt did not have the technical ability to recognize and fully understand the microorganisms responsible for these activities, they were able to understand the effects that bacteria can have upon food in terms of spoilage and fermentation. It is this understanding of bacterial effects that have led to some of our modern detection techniques.

Two main advantages of using bacterial cultures are reproducibility and the ability to separate bacteria for further cultures or testing. A bacterial culture consists of millions of individual bacterial cells that can be subdivided for future tests, including for antibiotic sensitivity. Culturing of bacteria in a clinical diagnostic laboratory consists of inoculating a patient sample into liquid or solid media. If bacteria are present in the sample, they will grow and divide rapidly until a suitable number are present for visual or biochemical identification. Once a culture grows to appropriate density, a smear can be prepared and examined using light microscopy and staining procedures to identify bacterial shape and arrangement.

In the clinical microbiology lab, Gram staining is one of the first steps in the culture based bacterial identification process. This process, developed by Hans Christian Gram in 1884, was the first advancement in bacterial identification [8]. Gram staining involves staining a bacterial smear with a crystal violet-iodine complex, rinsing with alcohol, and applying safranin counterstain. This method classifies bacteria based on cell shape, cell arrangement, and the biochemical nature of the bacterial cell wall.

Beyond conventional methods for bacterial detection and identification, there are newer methods such as the use of infrared spectroscopy [9] and those using flow cytometry [10] and particularly for use in enumeration of cells [11]. Other methods have used targets such as platelet concentrates, as they are relevant in bacterial spread from blood donations [12]. But to date, flow cytometry for bacterial cells has seen limited use due to the smaller size of prokaryotes and the limited DNA content. These limitations require greater sensitivity expense.

Bacteriophage also use the biochemical nature of the cell wall to discriminate bacterial targets [13,14]. Bacteriophage have evolved to discriminate bacteria rapidly and specifically using tail spikes and tail fibers that biochemically correspond to bacterial cell surface proteins. As different subspecies of bacteria have emerged, some with variations in antibiotic susceptibility, bacteriophages have adapted to infect these organisms. Importantly, bacteriophages will continue to evolve along with bacteria and these naturally occurring probes provide an innovative solution to the shifting sands of bacterial epidemiology.

We have used photoacoustics, or laser induced ultrasound, to generate acoustic responses in targets of interest for biosensing and other applications [15,16,17]. Photoacoustic flow cytometry has been used by several groups to detect circulating tumor cells (CTCs), both in vitro [18,19,20,21] and in vivo [22,23]. We have detected melanoma cells in blood samples by targeting their natural pigment with laser light, producing an acoustic signature that indicates the presence of the tumor cell. Additionally, we have used photoacoustic flow cytometry for the detection of bacterial contamination using bacteriophage [24].

For bacterial cells, the photoacoustic process is similar to melanoma cell detection with the added step of introducing bacteriophage to the sample prior to photoacoustic sensing. This method is similar to detecting nonpigmented CTCs [25]. Bacteriophage contain tail spike proteins that specifically target bacterial species, so that if we wanted to perform photoacoustic identification on Salmonella or one of its subspecies, we could select a bacteriophage, such as DET7, to tag the cell. In order to provide optical contrast to bacteriophage, such as that provided by the melanin granules in melanoma, we introduced Direct Red 81, a photostable protein dye capable of generating photoacoustic waves after laser irradiation to the bacteriophage suspension. Direct Red 81 molecules will attach to the side chains on bacteriophage proteins. These bonds are generally stronger than covalent bonds and result in a permanently dyed bacteriophage. By separating a blood sample into several subsamples with different dyed bacteriophages, photoacoustic generation will only occur in the subsample with the matched bacteriophage, thus identifying the bacterial species.

This process can be done using different bacteriophages. For example, each subsample would have a particular modified species of bacteriophage, such as the ones corresponding to *Staphylococcus aureus* (*Staph. aureus*), *Enterococcus faecium*, or *Pseudomonas aeruginosa*. If the original blood sample from the patient contained one of these bacterial species, only the subsample with the matched bacteriophage will attach to the bacterial cells, essentially painting the bacteria with red dye and enabling photoacoustic generation. Thus, the subsample that results in acoustic events in the photoacoustic flow cytometer identifies the infectious agent.

Adding Direct Red, a protein dye, to a suspension of bacteriophage results in a bacteria specific probe with optical absorption. In order to reduce variability in signal strength, we detached the bacteriophage capsids and attached dyed latex microspheres to them. This method had the added benefit of ridding the genetic material from the bacteriophage, which results in preventing normal lyses of the bacterial cells. Lysis of bacteria can take place in less than an hour after bacteriophage infection, so using bacteriophage tails with microspheres extends the time for clinical testing and imaging. By attaching bacteriophage tails to uniformly produced microspheres, we leverage the specificity of bacteriophage with the reproducibility and uniformity of commercially produced dyed microspheres. Dyed microspheres allow us to easily modulate color and signal strength as well as capture bound microspheres for further analysis.

Halting the lytic lifecycle is a crucial aspect of the process if this method will be used clinically for bacterial identification. The timeframe for the lytic lifecycle varies depending on the bacterial species and even on the specific bacteriophage [26]. The time depends on the doubling time of the bacteria and can be as short as 15 min. If native bacteriophage with Direct Red dye were used to tag bacteria for photoacoustic identification, then the process would have to be complete prior to lysis of the bacteria. By removing the capsids and, hence, the genetic material, lysis of the bacteria is avoided. Addition of dyed microspheres then provides the optical absorption necessary for photoacoustic generation.

We describe the process for separating capsids and subsequent attachment of red and blue latex microspheres. We then validate their use in photoacoustic flow cytometry under ideal conditions, enabling future studies where bacterial identification will be performed in other media, including urine and blood. Efficacy of photoacoustic detection in blood has been previously shown in our cancer studies, where CTCs were detected from the blood of cancer patients [19,21].

## 2. Materials and Methods

### 2.1. Photoacoustic Flow Cytometry

Our photoacoustic testing set up is based on our previously developed system for detection of circulating melanoma and bacterial cells. A schematic of the flow system is displayed below in Figure 1. Laser light at a wavelength of 532 nm and pulse repetition rate of 20 Hz was coupled into a 1000 μm optical fiber with numerical aperture 0.39 (Thorlabs, Newton, NJ, USA) and was used to deliver 5 ns laser pulses directly to our sample. Wavelength of 532 nm was chosen as it is a common laser line that has good absorption in our photoacoustic targets, including the microspheres used in this study. The short pulse duration of 5 ns is necessary to achieve stress confinement for producing robust photoacoustic waves. Laser energy was approximately 2 mJ per pulse out of the optical fiber.

For sample handling, a syringe pump system maintained a constant flow rate of 60 μL/min of the sample through the acoustic excitation chamber. The flow rate was set so that the pulse repetition rate of the laser and the flow chamber geometry would ensure irradiation of all material in the liquid sample. As this system is meant to detect relatively infrequent targets, such as CTCs or dilute suspensions of bacteria, we programmed a 1 s delay in the photoacoustic system after each detection event. This delay was to prevent double counting of suspended particles under flow. For a 1 mL sample, the run time was about 16 min, after a short delay caused by the sample traveling from the syringe pump to the flow chamber. The excitation chamber consisted of a quartz tube with a 400 μm outer diameter and 10 μm thick walls (Quartz 10 QZ, Charles Supper, Natick, MA, USA) submerged in Sonotech LithoClear acoustic gel (Next Medical Products Company, Branchburg, NJ, USA). The thin walled quartz was used to allow maximum acoustic energy to traverse the liquid/quartz boundary. The optical fiber was placed 5 mm from the quartz tube with a 2.25 MHz focused transducer (Olympus, Waltham, MA, USA) positioned below quartz tube. The transducer was used passively to detect photoacoustic events in the chamber. A 3D printed polylactic acid (PLA) was used to hold the quartz tube, optical fiber, and transducer with acoustic gel filling the internal volume.

To determine background noise the photoacoustic flow cytometry system was tested using phosphate buffered saline (PBS) and Phage Buffer (10 mM Tris, pH 7.5, 10 mM MgCl_2_, 68 mM NaCl). As a positive system control, 1 μm polystyrene microsphere (Polybead, Warrington, PA, USA) were tested and titered through the photoacoustic flow cytometry system at a concentration of $10^{7}$ beads per milliliter. For these microspheres, optical absorption is high enough to provide constant, robust signals to ensure the system is operating properly.

### 2.2. Production of Bacteriophage Tails

Bacteriophage was purified using cesium chloride (CsCl) gradient purification. Osmotic shock was used to removed bacteriophage capsids from tails as described earlier [27]. Bacteriophage were produced and concentrated to be $1\times 10^{12}$ pfu/mL and CsCl was added to increase the density of the solution to 1.5 g/mL to aid in sample separation. Bacteriophage was incubated in CsCl overnight allowing the CsCl to infuse into the bacteriophage DNA that was tightly packaged in the capsid. CsCl-infused bacteriophage were then rapidly diluted into phage buffer, causing the rapid diffusion of CsCl out of the bacteriophage. This rapid diffusion resulted in the separation of bacteriophage capsid and tails at the neck connecter. This process of separating intact capsids and tails has been used by bacteriophage biologists for many years [27,28]. Bacteriophage tails were further purified using Bio-Rad HPLC (Hercules, CA, USA) and purity was determined by spectrophotometry as well as electron microscopy. We used the BioCad Sprint ion exchange column and was used within normal flow rates by manufacturer’s recommendation. Protein purity and concentration was calculated from optical absorbance using a BioTek Synergy H1 (Winooski, VT, USA). A 96-well plate was used with the BioTek Synergy H1 to obtain multiple optical absorbance measurements at 260 nm and 280 nm. The Beer-Lambert law was then used to estimate the protein concentration. The absorbance ratio of the two wavelengths was then used to estimate the purity of the protein and any possible DNA contamination.

Electron micrographs were taken by Dr. James Conway in the Department of Structural Biology at the University of Pittsburgh. Micrographs were taken on a Thermo Fisher/FEI T12 Spirit using a Gatan US 1000 and Orius CCD camera (Hillsboro, OR, USA). Micrographs were examined to for the presence of contaminating DNA or groEL [29], both of which commonly purify with bacteriophage. Tail preparations were found to be of high quality and purity with no observation of contaminating DNA or groEL. Following confirmation of purity, tail preparations were used in later procedures for attachment to microspheres.

### 2.3. Attachment of Bacteriophage Tails to Streptavidin Coated Microspheres

Streptavidin coated dyed polystyrene microspheres with nominal diameter of 0.19 μm were obtained from Bangs Laboratories (Fishers, Indiana). Streptavidin coated microspheres were washed four times in PBS to remove stabilizer and antimicrobial agents used by the manufacturer. Microspheres were washed using PBS and using Spin-X concentrator columns three times. Biotin (Thermo Scientific EZ- Link Sulfo-NHS-Biotin, Thermo Fisher Scientific, Waltham, MA, USA) was prepared separately and resuspended in PBS at a concentration of at least 20 fold excess to the bacteriophage tail protein binding. Biotin and purified tails were combined and incubated on ice for two hours. After incubation, excess biotin was removed by dialysis using 2 kD molecular weight cut off dialysis cassettes (Slide-A-Lyzer, Thermo Fisher Scientific, Waltham, MA, USA). Biotinilated tails were incubated with washed streptavidin coated microspheres at room temperature for 30 min with gentle mixing. Microspheres were washed ten times to remove any unbound biotinylated bacteriophage tails. Microspheres with bound bacteriophage tails were then concentrated using slow speed centrifugation.

### 2.4. Photoacoustic Flow Cytometry Using Functionalized Microspheres

Overnight cultures of each *Staph. aureus* strain were prepared in Mannitol Salt Phenol Broth (MSB) media (Millipore Sigma, Burlington, MA, USA). Overnight cultures were diluted 1:20 and were regrown in lysogeny broth (LB) media for two hours to ensure synchronous cultures in exponential growth phase. Afterwards, re-growth cultures were pelleted and diluted to desired concentration for photoacoustic flow cytometry. Each sample was diluted to contain roughly 100 bacterial cells per test. Functionalized microspheres were added to diluted bacterial cultures and incubated at room temperature for 10 min to allow binding to bacterial surfaces for a total volume of about 1 mL. An excess of functionalized microspheres was added to each bacterial culture so that there were approximately 500 functionalized microspheres per bacterial cell. Combined samples were run on photoacoustic flow cytometry system rate of 60 μL/min as previously described [19,21,24].

In order to test our functionalized microspheres for binding and signal generation we used two bacterial strains. *Salmonella* LT2 is the target host for bacteriophage Det7 from which the tails were produced. Specificity of binding and host range of bacteriophage Det7 has previously been established [27]. As a negative binding control we used *E. coli* strain K12 to which bacteriophage Det7 does not bind. As mentioned above, black 1 μm polystyrene microspheres were tested in the photoacoustic flow cytometry system as a positive control at $10^{7}$ per milliliter and to give high detection signals. As a negative control, we first demonstrated that no signals were produced from our resuspension buffer PBS. Subsequently, both bacterial strains were run at $10^{7}$ per milliliter. We then tested streptavidin coated red and blue dyed 0.2 μm polystyrene microspheres. Each color microsphere was tested at $10^{8}$ and $10^{9}$ spheres per milliliter. This series of tests was to determine if free floating microspheres and untagged bacteria showed any photoacoustic generation.

To determine the binding and detection of our functionalized microspheres, we tested each combination of target and non-target bacteria with red and blue functionalized microspheres. Bacterial cultures of LT2 *Salmonella* and K12 *E. coli* were diluted into PBS and titered through the photoacoustic flow cytometry system at $10^{7}$ per milliliter. We ran three trials for each bacterial species for both red and blue microspheres.

## 3. Results

The absorption spectrum of blue and red microspheres is shown in Figure 2.

### 3.1. Verification of Functionalized Probes

We examined microspheres with attached tails using electron microscopy. Samples were negatively stained and multiple dilutions of microsphere with bound tails were examined. Uniform attachment of tails was observed. No microspheres were identified that did not have a full compliment of unbound bacteriophage tails. Very few free floating unbound bacteriophage tails, much less than 1%, were identified in each preparation suggesting that both our tail binding and washing procedures were effective. Additionally, we looked for signs of contamination by DNA or groEL. No presence of either was observed in any prepared samples suggesting a high level of purity. Examples of functionalized microspheres can be seen in Figure 3.

### 3.2. Controls

The control experiments are shown in Table 1. As expected, the negative controls showed no photoacoustic signals as they either lacked optical absorption for the cells or were too small and dilute in the case of the blue and red microsphers, while the positive control of 1 μm black microspheres showed 967 signals, which indicated essentially constant photoacoustic generation.

### 3.3. Bacterial Samples

The photoacoustic results of the tagged bacterial samples are shown in Table 2. As expected, there were multiple detections of targeted bacteria, *Salmonella* LT2, while the non-targeted bacteria, *E. coli* K12, showed very few or no detections. The average detections of *Salmonella* LT2 bacteria with red microspheres was 109.3±9.0. The average detections of *E. coli* K12 bacteria with red microspheres was 2.0±1.7. The average detections of *Salmonella* LT2 bacteria with blue microspheres was 86.7±13.2. The average detections of *E. coli* K12 bacteria with blue microspheres was 0.3±0.6.

## 4. Discussion

The CDC estimates nearly 2.8 million antibiotic resistant infections occur in the United States with over 35,000 deaths annually [30]. Early and rapid detection has been a goal for hospitals and governments alike. Using functionalized microspheres and photoacoustic flow cytometry we have expanded our detection capabilities of bacterial contaminants.

Functionalized microspheres were produced by attaching bacteriophage tails directly to streptavidin coated colored photo absorbing microspheres. The uniform size and color of the microspheres allows for a uniformity of signal. Additionally, the elimination of bacteriophage capsids and DNA allow us to remove the potential for lysis of bacterial cells. This inhibition of lysis enables the downstream capture and testing of detected bacterial cells.

In the positive control test of 1 μm black microspheres, we detected 967 photoacoustic events. This concentration of black microspheres was more than enough to generate detectable photoacoustic waves with each laser pulse. However, since each detection caused a one second interruption in counted detections, as explained above, there was a total of 1 detected event per second for the sixteen minutes of the sample run. These black microspheres were large enough to cause robust photoacoustic waves at $10^{7}$ per milliliter. However, the much smaller red and blue microspheres, even at higher concentrations, did not provide enough optical absorption to generate discernible acoustic waves. Even at $10^{9}$ per milliliter, assuming a uniform spatial distribution of microspheres, each individual microsphere would be separated 10 μm from other microspheres, which is a distance much greater than the size of a bacterial cell. With up to 500 red or blue microspheres with a diameter of 0.19 μm aggregated on a 1 μm cell, the composite volume is on the order of 2 μm${}^{3}$, which is similar to the volume of a detectable black microsphere at about half of a cubic micron.

We tested two colors of microspheres and binding to two different bacterial hosts. We assume that the free floating red and blue microspheres are too dilute to produce detectable photoacoustic events and would need to be approximately an order of magnitude more concentrated, based on the assumption that there are 10–20 bacteriophage attached to each cell, creating a high concentration of colored spheres.

Tails from bacteriophage Det7 were used to produce both red and blue functionalized microspheres. Det7 is a bacteriophage that exclusively binds to *Salmonella* bacterial species. As a non-target host we use *E. coli* K12 which Det7 does not bind to or infect. Both red and blue functionalized microspheres were tested in our photoacoustic flow cytometry system with *Salmonella* strain LT2. Red microspheres showed roughly 20% more detections with an average number of detections of 109.3 for red and 86.7 for blue microspheres. There are several possible reasons for this discrepancy in number of detections. First, the red microspheres absorb the green laser light slightly better than the blue microspheres of the same size. It is very likely that the difference in absorbance accounts for the roughly 20% difference in detections. It is also possible, though unlikely, that the dilutions of each bacterial culture were slightly skewed. Despite these differences in detections for target bacteria the number of detections for non-target bacteria are consistently zero indicating very good discrimination between bacterial strains and specificity of binding. Incidentally, detection of greater than 100 bacterial cells for trials 1 and 2 for red LT2 and for trial 3 of blue LT2 were probably due to the imprecision of adding bacterial cells to the sample.

Electron micrographs confirm complete coverage by bacteriophage tails to the surface of each microsphere. Bacteriophage tails were randomly oriented on the microsphere surface as confirmed by electron microscopy. Several other groups have shown that randomly oriented bacteriophage tails bind at nearly the same efficiency as oriented bacteriophage tails [31,32,33]. Therefore, we do not expect any significant reduction in binding capacity from the random orientation of tails as each microsphere has multiple tails attached allowing for multiple attachment points. While confirming the tail orientation a slight overabundance of tails was noticed. In future experiments we will try producing microspheres with a smaller number of attached tails to maximize the use of purified tails and minimizing any overcrowding issues.

Combining the host attachment specificity of bacteriophage tails and the uniform production and absorption of dyed polystyrene microspheres allows us to enumerate specific bacterial contaminants. This technique allows for the potential producing multi-target microspheres with any combination of binding produced by bacteriophage. The binding of bacteriophage is often to essential cell surface proteins making them far superior to antibody detection [34]. Additionally, bacteriophage attachment proteins have been shown to be some of the most stable protein structures discovered allowing ease of storage and long-term viability of functionalized microspheres as bacterial probes [35,36].

Photoacoustic flow cytometry in conjunction with functionalized microspheres presents a method of rapid bacterial detection and quantification with the added benefits of uniform signals and potential recovery of each detected bacterial cell. Further development of this method and combining this technique with our previous work on antibiotic sensitivities shows the potential for clinical applications and point of care use [37]. Rapid detection and identification of bacterial infection are not only a cost saving, but also more importantly potential lifesaving technology. Frequently the limiting factors for patient treatment is the time spent waiting for results. It is our hope that this technology can become the new gold standard and replace the 19th centrury technology of plate cultures and Gram staining.

Our future work centers on a more robust statistical validation of this method using a larger bank of target and non-target bacteria. We will also produce multi-host functionalized microspheres with wide target host ranges allowing for even more rapid identification of bacterial contamination.

## Figures and Tables

**Figure 1 micromachines-14-00573-f001:**
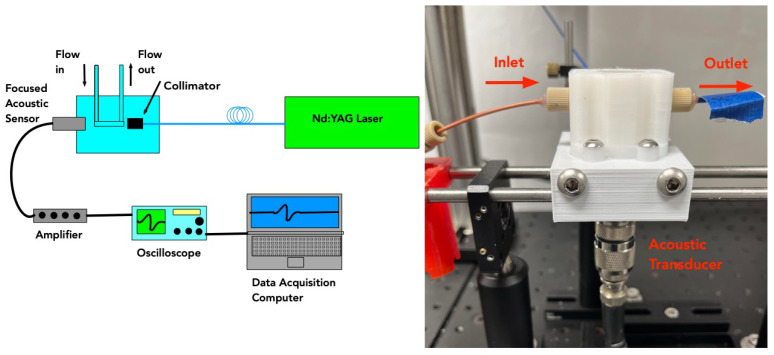
(**Left**) Schematic showing photoacoustic setup used for testing *Staph. aureus* samples. (**Right**) The actual flow chamber is shown.

**Figure 2 micromachines-14-00573-f002:**
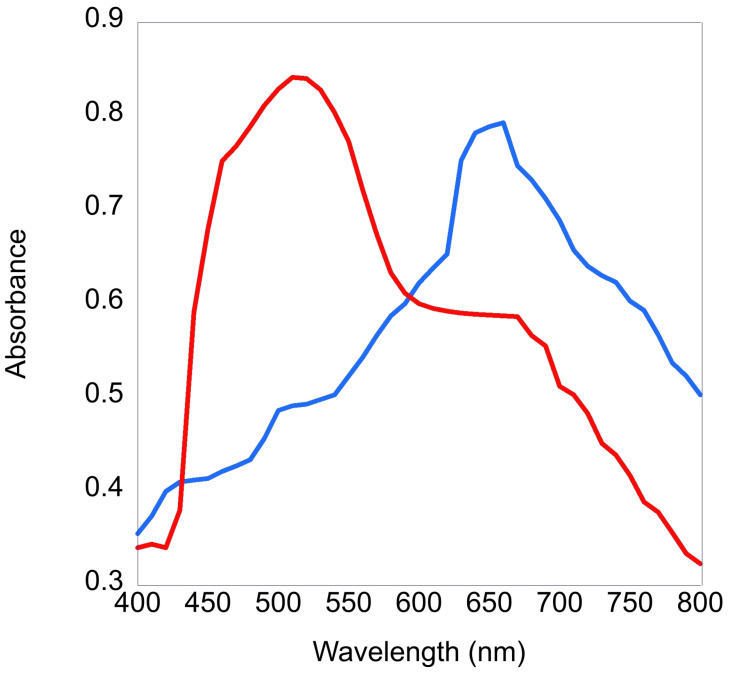
Absorbance of red and blue microspheres shown in red and blue traces, respectively.

**Figure 3 micromachines-14-00573-f003:**
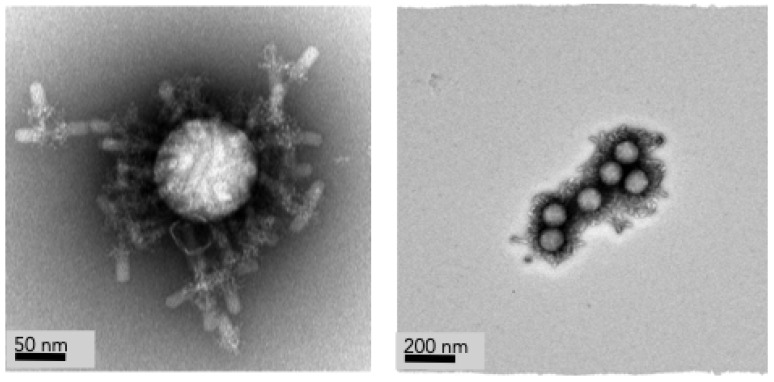
(**Left**) Single functionalized microsphere with multiple bound bacteriophage tails in random orientations. (**Right**) Multiple functionalized microspheres with attached bacteriophage tails. Micrographs take by Dr. James Conway, University of Pittsburgh.

**Table 1 micromachines-14-00573-t001:** Control experiments demonstrating positive signals from black microspheres while obtaining no signals from PBS, target and non-target bacteria, and red and blue microspheres.

Control Type	Sample	Detections
Positive	Black 1 μm microspheres	967
Negative	PBS	0
Negative	LT2 bacteria alone	0
Negative	K12 bacteria alone	0
Negative	$10^{8}$ red streptavidan 0.2 μm microsphere	0
Negative	$10^{9}$ red streptavidan 0.2 μm microsphere	0
Negative	$10^{8}$ blue streptavidan 0.2 μm microsphere	0
Negative	$10^{9}$ blue streptavidan 0.2 μm microsphere	0

**Table 2 micromachines-14-00573-t002:** Red functionalized microspheres (RFM) tested with target bacteria (*Salmonella* LT2) and non-target bacteria (*E.coli* K12). Blue functionalized microspheres tested with target bacteria (*Salmonella* LT2) and non-target bacteria (*E.coli* K12).

Microsphere	Bacterial Strain	Trial #1	Trial #2	Trial #3
Red	LT2	115	114	99
Red	K12	3	3	0
Blue	LT2	75	84	101
Blue	K12	0	0	1

## Data Availability

The data presented in this study is available on request from the corresponding author.

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
