# Peer review of "Photoacoustic Flow Cytometry Using Functionalized Microspheres for Selective Detection of Bacteria"

_micromachines, 2023, doi:10.3390/mi14030573_

Round 1

Reviewer 1 Report

This is a very interesting article described a unique technique of photoacoustic flow cytometry detection of bacteria using dyed tail fibers of bacteriophage.

I have some minor notes:

1)      Please, use a full species name of bacteria at first mention even for E. coli. For Salmonella, only the generic name was given. Species names should also be written in italics. I found different abbreviations for Staphylococcus aureus – Staph. aureus (line 62, 149) and S. aureus (In the caption to figure 1)

2)      In line 104 correction of “1x1012” is needed (should be “1x1012”)

3)      In line 110 some reference is missed in [?]

4)      Figure 2 does not clearly represent the scales.

5)      The sentences in lines 172-174 repeat sentences in lines 153-158.

6)      Authors hypothesized that this method could be usefull for the detection of bacterials in blood stream. But did authors test blood or others biological fluids as negative  control? Is this method able to detect bacteria not in clear buffer but in serum or plasma?

7)      Please add a relevant reference to “Further development of this methods and combining this technique with our previous work on antibiotic sensitivities shows the potential for clinical applications and point of care use.” (lines 236-238)

Reviewer 2 Report

Review micromachines-2036499 of

Photoacoustic flow cytometry using functionalized microspheres for selective detection of bacteria
by Robert H. Edgar, Anie-Pier Samson, Tori Kocsis, and John A. Viator
(submitted to Micromachines)

The manuscript at hand presents a method to detect Salmonella (LT2) bacteria in flow using photoacoustics. To be able to use the photoacoustic effect with bacteria, which technically are transparent to the used laser light, bacteriophages are functionalized with microbeads and attached to the bacteria.

The topic itself is highly relevant and the methodology of using bacteriophages as markers is innovative. Unfortunately, I fail to see the advancement over the already published work of the authors (10.1117/1.JBO.24.11.115003). Despite the huge overlap of the already published work, the manuscript at hand is not in a publishable state. The structure of the introduction is very confusing and, thus, hard to follow. Important concepts are not introduced, while other aspects, such as culturing of bacteria, are explained in detail and could be shortened substantially. Further, the introduction fails to give an overview of the state-of-the-art methods in the field and why the manuscript at hand is an improvement over already published work.             

 The Materials and Method section contains information that should be moved to the Result section and also lacks some essential justification of the chosen experimental parameters, such as the wavelength of the laser and the excitation frequency of the acoustic transducer.  

 The result section contains only a few sentences and two tables where it is not clear how the data have been obtained. Which signal has been used to detect the bacteria? What laser and piezo power? Which flow rate? What is the influence of the experimental parameters on the detection rate, and what is the working range? Is the demonstrated detection reasonable? For example, if 100 bacteria are injected, a detection of 115 does not make sense.         

According to the mentioned issues of the manuscript at hand, I must recommend rejection.

Reviewer 3 Report

This is a very interesting paper aimed detection of bacteria using beads functionalized with isolated bacteriophage tails as a contrast mechanism for photoacoustic flow cytometry.   Rapid identification of bacteria without the need for culture is clearly a problem of interest, and the method presented here appears to be sensitive and specific.  

However, there is not nearly enough detail provided about the detection method.  Based on looking up previous work from the same authors, I think what is happening here is that 1) the aqueous sample containing bacteria is partitioned into droplets by a co-flowing oil phase; 2) the limit of detection of the photoacoustic system is set such that a) isolated 0.2um diameter dyed beads are below the limit of detection at the baseline concentrations used, and at 10x that concentration; b) 0.2um red or blue beads agglomerated by binding between the phage tails on the beads and a bacterium are above the limit of detection; c) the 1um black beads used as positive controls are above the limit of detection even when isolated.  None of the things in this list is stated explicitly in the manuscript text, which will make it difficult for a new reader to understand the significance of the project. 

The authors should also add some more detail about their experimental methods and results, including:

*p3L110 - reference is missing for phage tail purification.  Without this reference or any quantitative information about the absorbance measurements or HPLC purification, it's hard to assess how successful this technique was.  Authors should provide data as well as the reference.

*p4L154: please provide volume of bacterial sample before and after addition of microspheres.  

*1x/10x microspheres used as controls: please provide an estimate of concentration in units of beads/uL, or an estimate for number of beads in PAFC detection volume assuming uniform distribution.  I am guessing that 1x is identical to the bead concentration used with the bacterial samples but this is not stated in the text.

* similarly, it would be very helpful to provide an estimate for the bead concentration detection limit for the red and blue beads.

There are also a number of typos, some of which are entertaining but none of which impair understanding: for example streptavidin in table 1, and p7L241 "golf standard" instead of "gold standard".  This is not a complete list.

NB on future work: it is not necessary for publication of this manuscript, but I would be very interested to see how this detection method fares in something closer to real-world samples spiked with bacteria of interest, rather than single types of bacteria in LB

Round 2

Reviewer 2 Report

Review micromachines-2036499 2nd round of

Photoacoustic flow cytometry using functionalized microspheres for selective detection of bacteria
by Robert H. Edgar, Anie-Pier Samson, Tori Kocsis, and John A. Viator
(submitted to Micromachines)

The authors thoroughly revised the manuscript and added most of the missing information.

If the editor thinks that there is not too much overlap with the already published work of the authors (10.1117/1.JBO.24.11.115003), I recommend accepting the paper after addressing the following minor revisions:

·         There are very few citations in the first paragraphs of the introduction. Please add more citations especially lines 16-20.

·         Further, there is still only very little discussion about state of the art in flow bacteria detection. There should be more discussion about other methods and the advantages of the presented method

·         The authors added the used flow rate as suggested. However, a small section that discusses the used flow rate would be nice. What is the flow rate limit of the used detection method? Is the presented flow rate clinically relevant?

·         What is the inner and outer diameter of the quartz tube? Just wall thickness is mentioned.

·         The driving voltage/power of the piezoelectric element is still missing. Did the authors use a power amplifier?

·         Line 144: “to increase the density of the solution to 1.5 g/ml.” Why has the density been increased?

·         Line 150: “purified using Bio-Rad HPLC (Hercules, California)”. There should be more information added regarding the HPLC system. What column has been used? What flow rates?

·         Line 216: “bacteriophage tails. Very few free floating” Very few is very unspecific. How much %?

·         Line 236-239 should appear in the intro and can be removed in the discussion

Author Response

There are very few citations in the first paragraphs of the introduction. Please add more citations especially lines 16-20. - Several new citations are in the document, particularly in the Introduction.

  • Further, there is still only very little discussion about state of the art in flow bacteria detection. There should be more discussion about other methods and the advantages of the presented method - A paragraph has been added, along with citations, on other methods, including flow, and their shortcomings, which contrast with our method, introduced later in the Introduction.
  • The authors added the used flow rate as suggested. However, a small section that discusses the used flow rate would be nice. What is the flow rate limit of the used detection method? Is the presented flow rate clinically relevant? Text has been added and other material, in response to the other reviewer, include an analysis of the flow rate, total time, and control detections.
  • What is the inner and outer diameter of the quartz tube? Just wall thickness is mentioned. Outside diameter has been added.  ID can be deduce from that and wall thickness.
  • The driving voltage/power of the piezoelectric element is still missing. Did the authors use a power amplifier? The piezoelectric element was only used passively as a detector and was not driven.  A sentence was included to note this.
  • Line 144: “to increase the density of the solution to 1.5 g/ml.” Why has the density been increased? A phrase was added to indicate the use of CsCl for sample separation.
  • Line 150: “purified using Bio-Rad HPLC (Hercules, California)”. There should be more information added regarding the HPLC system. What column has been used? What flow rates? Additional information was included on the HPLC.  Unfortunately, we did not record flow rate, but we were certain we stayed within recommended limits, so we added additional verbiage to show that.
  • Line 216: “bacteriophage tails. Very few free floating” Very few is very unspecific. How much %?  Unfortunately, we did not do a comprehensive data analysis of free floating tails, as we were sure that the ultimate photoacoustic results would show efficacy.  Without keeping a larger number of electron micrographs, we can't be more specific on a percentage, other than to indicate in the text that the free floating tails was much less than a percent.  We regret not being able to provide more precise information.
  • Line 236-239 should appear in the intro and can be removed in the discussion. This section was removed.

Reviewer 3 Report

This is greatly improved in clarity of presentation.  I have a few remaining points:

1) Please provide some quantitative estimate for the phage tail purification.  Authors describe methods but not results or what quantitative criteria they are using to measure successful purification.

2) Taken together, the concentrations, volume, and detection limits given are hard to reconcile.  Part of the difficulty is that the detection volume, including the interior dimensions of the quartz cell, are not stated (but should be).  I think the paper probably needs one more close pass , checking that the experimental conditions are accurate and clearly stated for controls and for the sample measurements.  In particular:

a) the concentrations stated for control measurements are 1e7/mL for the positive control 1um microspheres, and 1e8 (blue?) and 1e9 (red?)/ mL  for the colored microspheres, in a 1mL volume.  But the positive control only shows ~1k detection events.  Is this the same total measurement time as in the sample runs?  Do the 967 events in the positive controls correspond to the full mL volume run at 60uL/min?

b) the concentration stated/implied for the experiments is 100 bacterial cells and 500x100=50000 colored beads per mL (line 188), and the data in table 2 showing results suggests that the full mL of sample volume was run (at 60uL /min, so ~16 minutes). Is this correct?  It's a little surprising to me that the unfunctionalized beads generated no signal despite being run at 20000x (=1e9/50000) greater concentration in the control run.  I understand that the functionalized beads will aggregate on the bacteria, but if you're only getting 967 detection events from 1e7 black beads run in the same time frame, this suggests that the effective detection volume could be fairly large - or that the detection method is fairly sensitive to the distance between the aggregated particles.  (and also suggests that the time needed per measurement point is fairly large, ~1s.)  Please clarify. 

Author Response

1) Please provide some quantitative estimate for the phage tail purification.  Authors describe methods but not results or what quantitative criteria they are using to measure successful purification. Unfortunately, we did not do a comprehensive data analysis of free floating tails, as we were sure that the ultimate photoacoustic results would show efficacy.  Without keeping a larger number of electron micrographs, we can't be more specific on a percentage, other than to indicate in the text that the free floating tails was much less than a percent.  We regret not being able to provide more precise information.

2) Taken together, the concentrations, volume, and detection limits given are hard to reconcile.  Part of the difficulty is that the detection volume, including the interior dimensions of the quartz cell, are not stated (but should be).  I think the paper probably needs one more close pass , checking that the experimental conditions are accurate and clearly stated for controls and for the sample measurements.  In particular:

a) the concentrations stated for control measurements are 1e7/mL for the positive control 1um microspheres, and 1e8 (blue?) and 1e9 (red?)/ mL  for the colored microspheres, in a 1mL volume.  But the positive control only shows ~1k detection events.  Is this the same total measurement time as in the sample runs?  Do the 967 events in the positive controls correspond to the full mL volume run at 60uL/min?  More information was included in the Material and Methods and in the Discussion to explain this.  One fact we included is that the detection software has a one second delay to prevent double counting of particles.  Since this system is meant to detect relatively rare events, a single particle caught in an eddy or subject to subsequent laser pulses can be counted twice or more, thus the one second suspension of particle counting was included in the system.  Thus, for the high concentration of black microspheres, 967 counts corresponds to one a second for 16 mintutes (960).

b) the concentration stated/implied for the experiments is 100 bacterial cells and 500x100=50000 colored beads per mL (line 188), and the data in table 2 showing results suggests that the full mL of sample volume was run (at 60uL /min, so ~16 minutes). Is this correct?  It's a little surprising to me that the unfunctionalized beads generated no signal despite being run at 20000x (=1e9/50000) greater concentration in the control run.  I understand that the functionalized beads will aggregate on the bacteria, but if you're only getting 967 detection events from 1e7 black beads run in the same time frame, this suggests that the effective detection volume could be fairly large - or that the detection method is fairly sensitive to the distance between the aggregated particles.  (and also suggests that the time needed per measurement point is fairly large, ~1s.)  Please clarify.  Blue and red microspheres are much smaller, with the black microspheres about 150 greater in volume and, hence, optical absorption.  We include some comments in discussion that show how free floating microspheres at that smaller volume would not create detectable photoacoustic events, but that they, when aggregated in the expected numbers on a 1 micron cell, are comparable in volume and optical absorption to black microspheres.